# When Data Is Scarce: The Strength of the Prior in Tabular Foundation Models

Florian D. van Leeuwen [1]    Sara van Erp [1]

## Abstract

We present a systematic evaluation of Prior-Data Fitted Networks (PFNs) in small-sample ($n < 500$) prediction tasks. First, through synthetic experiments, we quantify how informative parameter priors must be for correctly specified parametric models to match PFN predictive performance. Our results indicate that PFNs are competitive even against models with strong, well-calibrated priors. Second, using subsamples of the TabArena benchmark, we show that PFNs outperform traditional regression and tree-based methods across both classification and regression tasks in small-sample settings.

## 1. Introduction

Sample size limitations pose a serious constraint for accurate predictive modeling. In many fields, such as medicine (Wynants et al., 2020; Varoquaux & Cheplygina, 2022; Tsegaye et al., 2025; Zhong et al., 2025), economics (Carriero et al., 2019), and social sciences (Vankelecom et al., 2025), there are constraints on the sample size due to factors such as the cost of high-quality data collection and small populations. With small samples, flexible models might be particularly prone to overfitting on the training data, leading to poor generalization (Bias-variance trade-off; Bishop, 2006). To combat this, researchers commonly resort to "simpler" methods, such as linear or logistic regression (Hillel et al., 2021; Dhiman et al., 2022). While low in variance, the bias of such linear prediction functions may be large, still leading to poor predictions. Alternatively, tree-based models, such as XGBoost and CatBoost, work very well for tabular data (Grinsztajn et al., 2022; Shwartz-Ziv & Armon, 2022; McElfresh et al., 2023). However, adequate tuning of hyperparameters also requires sufficiently large sample sizes (Hastie et al., 2009).

When faced with a small sample size, an alternative approach is to encode domain knowledge into the prediction model through a Bayesian framework. By placing reasonable priors on the model parameters, the model variance can be reduced. Simultaneously, the bias may be small if the prior distributions are appropriate for the given problem. However, when the model parameters are not easily interpretable (e.g., the weights of neural networks), translating prior information into prior distributions can be challenging. While general shrinkage priors exist to induce sparsity in the model (Carvalho et al., 2010; Piironen & Vehtari, 2017; van Erp et al., 2019), they are particularly well suited when many parameters are expected to be negligible or only weakly associated with the outcome, but they are not intended to encode strong, parameter-specific information about effect sizes.

Recently, tabular foundation models in the form of Prior-Data Fitted Networks (PFNs) have been introduced (Müller et al., 2022). In contrast to traditional Bayesian models, the prior is not specified explicitly in a parametric form, but is defined through a generative process and learned from a large number of synthetic datasets. PFNs are trained to directly estimate the posterior predictive distribution (PPD) for new samples. A researcher will use the trained model and thus has no direct influence over the prior. PFNs have surged in popularity due to SOTA performance across different domains (Erickson et al., 2025; Hollmann et al., 2025; Qu et al., 2025; Ye et al., 2025). However, most evaluations consider datasets with over 500 records, which can still be considered large in many disciplines. It remains unclear how these PFNs perform with small sample sizes ($n < 500$).

**Our contributions are:**

- Assessing PFN predictive performance relative to correctly specified Bayesian models with priors of varying informativeness, across different sample sizes.
- Empirically evaluating PFN performance in small-sample settings by subsampling datasets from the TabArena benchmark.

## 2. Related Work

Müller et al. (2022) evaluate the performance of PFNs against Gaussian Processes as a function of the number

[1]Department of Methodology and Statistics, Utrecht University, Utrecht, The Netherlands. Correspondence to: Florian D. van Leeuwen <f.d.vanleeuwen@uu.nl>.

*Proceedings of the $2^{nd}$ ICML Workshop on Foundation Models for Structured Data*, Seoul, South Korea. 2026. Copyright 2026 by the author(s).

of in-context samples. In synthetic experiments, they show that when a PFN is trained on many datasets drawn from the correct prior (i.e., matching the data-generating process) and provided with a sufficient number of in-context examples, its performance can approach that of the exact Gaussian Process posterior.

To date, only one study has systematically evaluated PFNs on empirical small-sample datasets (Knauer et al., 2024). Their benchmark consisted of 44 binary classification datasets with fewer than 500 samples stemming from the larger PMLB collection (Romano et al., 2022). Knauer et al. (2024) showed promising results on the performance of a previous TabPFN variant, v0.1.9 (Hollmann et al., 2023), on small data sets. However, the authors note that part of the evaluation data was used to finetune the respective TabPFN model, potentially overestimating model performance.

## 3. The Role of the Prior

In a classical Bayesian analysis, the posterior is proportional to the product of a likelihood and a prior:

$$p(\theta|D) \propto p(D|\theta) \cdot p(\theta), \quad (1)$$

where $\theta$ denotes the model parameters, $D$ is the data, $p(D|\theta)$ is the likelihood of the data, $p(\theta)$ is the prior and $p(\theta|D)$ is the posterior. In a prediction context, the (training) data consists of pairs of features and outcomes: $D = \{(x_i, y_i)\}_{i=1}^n$, with $n$ being the sample size. The posterior predictive distribution (PPD) can then be derived for a new sample $x_{new}$:

$$p(y_{new}|x_{new}, D) = \int p(y_{new}|x_{new}, \theta)p(\theta|D)d\theta \quad (2)$$

Often, samples are taken from the posterior $p(\theta|D)$ using some MCMC method, and these can then be combined with a new sample to obtain a Monte Carlo estimate of the PPD. Assuming the data are fixed, the predictive power of a Bayesian model is affected by the prior and the likelihood. While asymptotically ($n \rightarrow \infty$) the role of the prior becomes irrelevant [1], in the small sample regime, the prior can exert substantial influence on the resulting PPD (Gelman et al., 2013).

PFNs are direct approximators [2] of the PPD. Contrary to the prior in a parametric Bayesian model, which is over the parameters, this prior is directly over the prediction task $t$ itself (the mapping from X to Y) (Müller et al., 2022). The PPD is then given as:

---

[1]Assuming the prior has support wherever the posterior has.

[2]Note that approximating the PPD is distinct from methods like variational inference (Blei et al., 2017), which approximate the posterior of the parameters.

$$p(y_{new}|x_{new}, D) = \int p(y_{new}|x_{new}, t)p(t|D)dt \quad (3)$$

Compared to Equation 2, the weights of the model ($\theta$) do not obtain a posterior distribution. PFNs are trained by minimizing the Prior-Data Negative Log-Likelihood (i.e., the cross-entropy on hold-out samples from the prior synthetic datasets; Müller et al., 2022; Hollmann et al., 2023):

$$L_{PFN} = \mathbb{E}_{((x_{new}, y_{new}) \cup D) \sim p(D)}[-\log q_\theta(y_{new}|x_{new}, D)]. \quad (4)$$

Minimizing this loss results in a Kullback-Leibler optimal approximation of the true PPD (Müller et al., 2022; Nagler, 2023). After being trained, PFNs use in-context learning to relate the data (X, Y) to a task $t$ in the prior, and use this mapping to approximate the PPD for new samples. PFNs even have the ability to produce prediction functions that are not directly seen in the prior (Müller et al., 2024). Specifying the prior, $p(D)$, as a data-generating mechanism offers much flexibility, as the only restriction for training a PFN is to be able to sample from the prior (Müller et al., 2025). The downside is that the computational cost of PFNs is in the training phase; many synthetic datasets are sampled from the prior for the model to generalise well. The benefit comes at inference time, as the same PFN can be used for many different prediction problems.

## 4. Experiments

We evaluate PFNs in two settings: (1) synthetic experiments comparing PFNs to correctly specified models with and without informative priors, and (2) subsampled datasets from the TabArena benchmark (Erickson et al., 2025) to assess small-sample empirical performance. We evaluate four PFN models, TabPFN v2.5, TabICLv2, Real-TabPFN v2.5, and TabDPT v1.1 (Grinsztajn et al., 2026; Qu et al., 2026; Garg et al., 2025; Ma et al., 2026), where the latter two use real data in their training regime, against XGBoost, Cat-Boost, RealMLP (Chen & Guestrin, 2016; Prokhorenkova et al., 2018; Holzmüller et al., 2024), and linear/logistic regression as baselines. Details on the model implementations can be found in App. A.

### 4.1. Synthetic Data

Our synthetic experiments mimic an ideal world in which a researcher specifies the correct likelihood for the data at hand, including a correct mapping from input to output. We simulate data with normally distributed predictors $X \sim N(\mu, \Sigma)$ and continuous and binary outcomes. For continuous outcomes, $y = f(X) + \epsilon$, and for binary

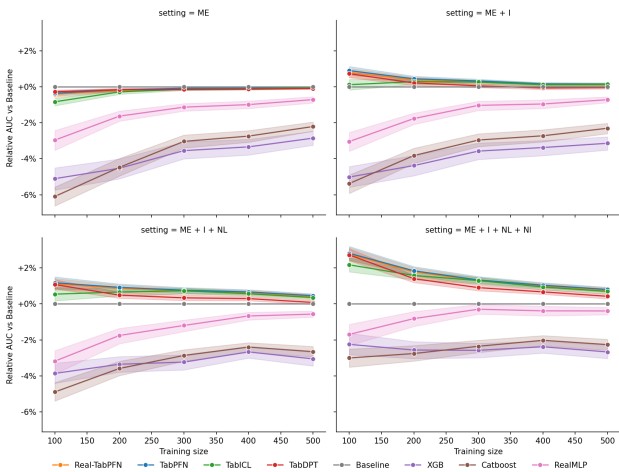

*Figure 1.* Relative AUC to a correctly specified baseline model, aggregated over the number of predictors. Positive values indicate better performance. The error bars indicate the 95% confidence intervals around the mean.

outcomes $y \sim \text{Bernoulli}(logit^{-1}(f(X) + \epsilon))$, both with $\epsilon \sim N(0, \sigma^2)$ [3]. See App. B for more details. In all settings, we compare against a baseline model where the functional form is correctly specified, but the model parameters are estimated from the data. In harder settings, the baseline model is prone to high variance due to its many parameters.

**Results.** Figure 1 shows that for binary classification, PFNs are on par or slightly better than the correctly specified model with uninformative priors on the parameters. This result holds for AUC and for the Brier score (Figure 5, App. C). For regression, the MSE for the tested models was larger compared to the baseline model (Figure 6, App. C). Only in the most difficult scenario (ME + I + NL + NI) with 100 training samples did some PFNs come within 1% of the baseline model performance. This worse performance, compared to the classification case, might relate to the output being a full distribution instead of a single estimate. Tab-DPT performed worse than the other PFNs, and in some cases, it performed worse than RealMLP.

The uncertainty estimates from the predicted distribution for regression tasks indicate that TabPFN/Real-TabPFN have slightly too narrow prediction intervals, while TabICL has slightly too large prediction intervals (Figure 7, App. C). Most deviations, however, fall within 1% of the nominal coverage rate.

In some cases, researchers might have domain knowledge

---

[3]The function f(X) is defined with increasing complexity: starting from a linear model with main effects (ME), then adding first-order interactions (ME + I), then also adding nonlinear transformations of the predictors (ME + I + NL), and finally extending the model with additional non-informative predictors (ME + I + NL + NI).

about a prediction problem. In that case, we could infer some knowledge about the parameters $\theta$ by setting informative priors. To assess how such a prior interplays with the PFNs, the focus is narrowed towards binary classification, where the PFNs outperformed the baseline model. The setting = ME + I + NL is chosen to have a challenging prediction function, with the baseline method having the model with the fewest number of parameters possible.

A Bayesian model is fit with priors based on a Gaussian distribution, $\theta_i \sim N(\mu_i, \sigma_j^2)$. The values for $\sigma_j$ can then be varied to assess how strong the priors need to be to achieve similar or better predictive performance compared to PFNs. Lower values for $\sigma_j$ indicate stronger prior knowledge, as the prior will be more peaked around the correct mean. Figure 2 shows the AUC values for PFNs compared to the baseline model with informative priors over the parameters. It can be seen that the PFN's performance increases as the sample size increases, and that at 500 samples, the performance becomes close to the Bayesian model with very strong (correct) prior knowledge on the parameters. In this example, PFNs with 300 samples for in-context learning already have comparable predictive performance compared to a parametric model with the correct functional form and strong ($\sigma = 1$) informative priors.

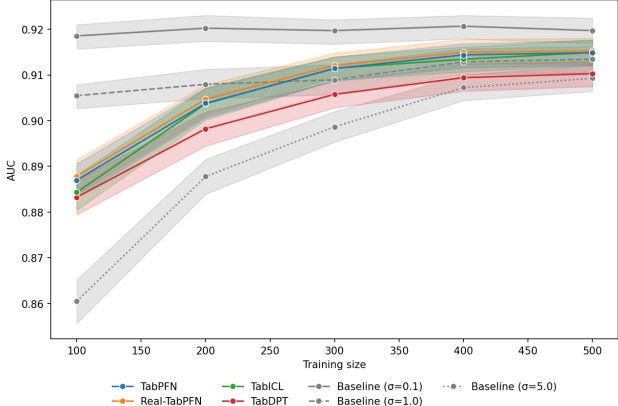

*Figure 2.* The absolute AUC values for different PFNs and Bayesian regression models for the ME + I + NL setting with 6 predictors. For the Bayesian models, the priors on all regression coefficients are informative, except the intercept, which has an uninformative prior. The informative priors all match the coefficient used in the data-generating mechanism; only the variance parameter of the prior distributions varies. The error bars indicate the 95% confidence intervals around the mean.

### 4.2. Empirical Data

Rather than using small datasets as in Knauer et al. (2024), we subsample the training set from larger datasets, enabling better out-of-sample estimation with a large test set (500 samples). Our empirical datasets stem from the TabArena benchmark (Erickson et al., 2025), restricted to binary clas-

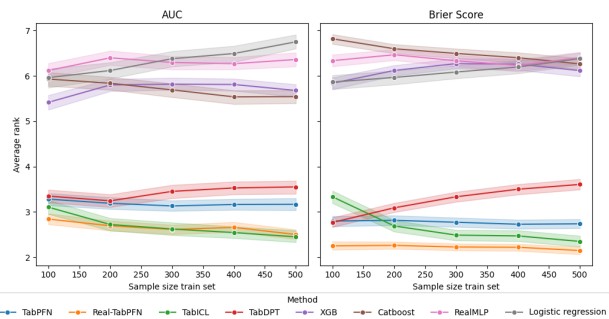

*Figure 3.* The average rank based on AUC and Brier score by training size, over all datasets used, lower values indicate better performance. The error bars indicate the 95% confidence intervals around the mean.

*Figure 4.* The average rank for the MSE (lower is better) and the average non-ranked coverage rates for the methods that provide uncertainty intervals over all datasets used. For the prediction intervals, an $\alpha$ of 0.05 was used; the dashed line indicates perfect coverage. The error bars indicate the 95% confidence intervals around the mean.

sification and regression tasks. We exclude datasets with more than 100 predictors, to avoid high-dimensional subsamples ($p > n$), and datasets with missing values (since not all models handle these natively), leaving 19 classification and 11 regression datasets. Models are evaluated by ranking their performance metric across subsamples. More details and the meta-data of the datasets can be found in App. D. No correctly specified baseline exists, as the data-generating mechanism is unknown. The baseline model is replaced by a Linear/Logistic regression model with only main effects.

**Results.** For binary classification tasks, there is a clear divide in performance between PFNs and other models (Figure 3, Table 13). We observe that Real-TabPFN/TabICL have the best performance based on AUC. Real-TabPFN outperforms TabICL on the Brier score, although TabICL seems to improve with an increased number of samples.

For regression tasks, we see that PFNs outperform the other methods by a wide margin in terms of the average rank (Figure 4, Table 14). The TabICL model has the lowest MSE across the experiments and seems to become better as the sample size increases. For the selection of models that output uncertainty estimates, the coverage of the prediction interval was calculated. All PFNs show small undercoverage and thus provide slightly overconfident uncertainty estimates. Interestingly enough, this effect seems stable over the sample size for the (Real-)TabPFN models while the undercoverage decreases for TabICL as $n \to 500$.

Averaged over all the sub-sample sizes, Real-TabPFN/TabICL perform the best for binary classification on AUC and Real-TabPFN solely based on Brier score (Table 15, App. E). TabICL performs the best for the regression tasks based on MSE (Table 16, App. E). The averaged results by dataset can be found in Table 17/18 App. E.

## 5. Conclusion

In this work, we show that PFNs perform particularly well in small-sample settings. Based on synthetic simulations, we find that, for classification tasks, PFNs approach the performance of a correctly specified model equipped with strongly informative priors on its parameters. This suggests that even when the true functional form is known and substantial prior knowledge is available, matching the predictive performance of PFNs remains challenging.

In empirical evaluations, we further observe strong performance of PFNs across datasets. In particular, Real-TabPFN/TabICL perform best in classification tasks, while TabICL shows the strongest results in regression settings. Notably, TabICL achieves this while being trained exclusively on synthetic data. A limitation of the empirical comparison is that the linear and logistic regression models only include main effects. In practice, researchers with domain expertise might want to include a targeted set of non-linear relations, possibly in combination with shrinkage priors, to boost predictive performance. While RealMLP and the tree-based methods were not exhaustively tuned to control computational cost, supplementary experiments with reduced tuning yielded broadly consistent results.

A natural direction for improvement is to incorporate domain knowledge, which parametric models would use to derive informative priors, into PFNs. Recent work from Reuter et al. (2026) demonstrates that this is feasible in a causal setting, where PFNs can be conditioned on partial graph structures to improve predictive performance. Extending this idea, one natural next step would be to develop a user-friendly mechanism for incorporating prior information into PFNs for general predictive tasks.

## Acknowledgements

We would like to thank Thom Volker and Alex Carriero for helpful discussions and comments on an earlier draft.

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

## A. Model Specifications and Implementation Details

The details for the models that are used in this paper can be found in Table 1/ 2. For most models, the default settings were used. RealMLP, XGBoost and CatBoost are optimized using cross entropy for classification and mse for regression tasks.

Additionally, for the experiment with informative priors, Bayesian regression models are fit using *bambi* (Capretto et al., 2022), which relies on *pymc* (Abril-Pla et al., 2023). The default sampler is used, which is a Hamiltonian Monte Carlo sampler, specifically the No-U-Turn Sampler algorithm (Hoffman & Gelman, 2014). For each model, 4 chains are used with 2000 draws warmup, 2000 for sampling. For the regression parameters, a normal prior is set with the correct mean and a different variance based on the setting. An uninformative prior $N(0, 10^2)$ is placed on the intercept. For other terms, default parameters were used, which are described here (Westfall, 2017).

*Table 1.* Overview of classification methods, corresponding functions, packages, and hyperparameters.

| Method | Function | Package | Hyperparameters |
|---|---|---|---|
| TabPFN v2.5 | TabPFNClassifier[a] | tabpfn | default |
| Real-TabPFN v2.5 | TabPFNClassifier[b] | tabpfn | default |
| TabICLv2 | TabICLClassifier | tabicl | default |
| TabDPT v1.1 | TabDPTClassifier | tabdpt | default |
| RealMLP | RealMLP_TD_Classifier | pytabkit | {n_cv=2/5[c], n_refit=1} |
| XGB | XGB_TD_Classifier | pytabkit | {n_cv=5, n_refit=1} |
| CatBoost | CatBoost_TD_Classifier | pytabkit | {n_cv=5, n_refit=1} |
| Logistic regression (Baseline) | LogisticRegression | sklearn.linear_model | default |
| Bayesian Logistic regression (Baseline) | Model | bambi | default |

[a] Checkpoint is tabpfn-v2.5-classifier-v2.5_default-2.ckpt.
[b] Checkpoint is tabpfn-v2.5-classifier-v2.5_default.ckpt.
[c] For the empirical datasets n_cv was set to 2 to reduce some of the computational cost.

For the PFNs with uncertainty estimates, first a model was fit to obtain the mean predicted value, and then a model was fit to obtain the quantiles. This was done to prevent some models from outputting the median for the prediction and other models from outputting the mean.

*Table 2.* Overview of regression methods, corresponding functions, packages, and hyperparameters.

| Method | Function | Package | Hyperparameters |
|---|---|---|---|
| TabPFN v2.5 | TabPFNRegressor[a] | tabpfn | default |
| Real-TabPFN v2.5 | TabPFNRegressor[b] | tabpfn | default |
| TabICLv2 | TabICLRegressor | tabicl | default |
| TabDPT v1.1 | TabDPTRegressor | tabdpt | default |
| RealMLP | RealMLP_TD_Regressor | pytabkit | {n_cv=2/5[c], n_refit=1} |
| XGB | XGB_TD_Regressor | pytabkit | {n_cv=5, n_refit=1} |
| CatBoost | CatBoost_TD_Regressor | pytabkit | {n_cv=5, n_refit=1} |
| Linear regression (Baseline) | OLS (sm.OLS) | statsmodels | default |

[a] Checkpoint is tabpfn-v2.5-regressor-v2.5_default.ckpt.
[b] Checkpoint is tabpfn-v2.5-regressor-v2.5_real.ckpt.
[c] For the empirical datasets n_cv was set to 2 to reduce some of the computational cost.

# B. Synthetic experiments setup

In the simulation, the aim is to assess the performance of tabular prediction models compared to a correctly specified linear model. In essence, the features are transformed to fit the data-generating mechanism, but the effect sizes still need to be estimated. Using such a setup allows us to study whether a model with prior knowledge, e.g. PFNs, can pick up the correct functional form only on the non-transformed features. Four scenarios are studied with increasingly difficult prediction problems:

- ME: Main effects only
- ME + I: Main effects and interactions
- ME + I + NL: Main effects, interactions, and non-linear effects
- ME + I + NL + NI: Main effects, interactions, and non-linear effects, Non-informative predictors

The X values are generated from a multivariate normal distribution:

$$X \sim N(\mu, \Sigma)$$

with $\mu$ being a vector of zeros and the covariance matrix $\Sigma$ where all off-diagonal values are equal to a set value $\rho$.

**Main effects (ME):**

$$y = g(\beta_0 + \sum_{j=1}^{p} \beta_j X_j + \epsilon),$$

with

$$g(x) = \begin{cases} x, & \text{if task is regression,} \\ \text{Bernoulli}(\frac{1}{(1+\exp(-x))}) & \text{if task is classification.} \end{cases}$$

and $\epsilon \sim N(0, \sigma^2)$.

**Main effects + Interactions (ME+I):**

$$y = g(\beta_0 + \sum_{j=1}^{p} \beta_j X_j + \sum_{j<k} \gamma_{jk} X_j X_k + \epsilon)$$

**Main effects + Interactions + Non-linear functions (ME+I+NL)**

$$y = g(\beta_0 + \sum_{j=1}^{p} \beta_j X_j + \sum_{j<k} \gamma_{jk} X_j X_k + \sum_{j=1}^{p} \delta_j \phi_j(X_j) + \epsilon)$$

**Main effects + Interactions + Non-linear functions + Non-informative params (ME+I+NL+NI)**

The function for $y$ stays the same, but p predictors ($X_{noise}$) are added, which are drawn from a Normal distribution with no correlation between any of the predictors or the outcome. The non-baseline models will only see the "raw" $X$ values, whereas the baseline model will obtain the $X_{transform}$ values, which vary depending on the setting (Table 3).

*Table 3.* The transformed values of X used for the baseline model to match the prediction function.

| Setting | $X_{transform}$ | # predictors |
|---|---|---|
| ME | $[X]$ | $p$ |
| ME + I | $[X, X_j X_k]$ | $p + \binom{p}{2}$ |
| ME + I + NL | $[X, X_j X_k, \phi_j(X_j)]$ | $p + \binom{p}{2} + p$ |
| ME + I + NL + NI | $[X, X_j X_k, \phi_j(X_j), X_{noise}]$ | $p + \binom{p}{2} + p + p$ |

The effect sizes are defined in table 4. The residual SD is set to $\sigma = 5$ for all experiments.

*Table 4.* Effect sizes and non-linear basis functions used in the four data-generating scenarios.

| Term | Effect size | Function $\phi_j$ |
|---|---|---|
| $\beta_j X_j$ | $\beta_j = j$ | – |
| $\gamma_{jk} X_j X_k$ | $\gamma_{jk} = 1$ | – |
| $\delta_j \phi_j(X_j)$ | $\delta_j = 1$ | $\phi_j \sim \text{Uniform}\{\, x^2,\ \log|x + 1|,\ |x|,\ e^x,\ \sqrt{|x|}\,\}$ |

The simulation criteria are chosen to cover a wide range of small sample datasets, as can be seen in Table 5.

*Table 5.* Simulation conditions

| Parameter | Conditions |
|---|---|
| Setting | {ME, ME + I, ME + I + NL, ME + I + NL + NI} |
| Train sample size | {100, 200, 300, 400, 500} |
| p | {2, 6, 10} |
| outcome | {binary, continuous} |

The test set is fixed to 500 cases to adequately estimate the performance of the predictive models. For all 120 scenarios, 100 datasets were simulated. All methods were evaluated on these datasets, except for the models which rely on heavy tuning (RealMLP, XGB and CatBoost), for these models only 50 datasets were used.

To reduce the computation cost, frequentist regression models were used for the cases where there were uninformative priors on the parameters. For the uncertainty estimates for the prediction interval of the frequentist linear regression model, the following formula is used:

$$\hat{y}_{new} \pm t_{\alpha/2, n-p} \sqrt{\sigma^2 (1 + x_{n+1} (X^T X)^{-1} x_{new}^T)},$$

with $X$ the training data, $x_{new}$ a test sample and $\sigma^2$ being the estimated residual error.

## C. Synthetic experiments additional results

The raw results on which Figure 1 is based are given in Table 6.

*Table 6.* AUC scores across experimental settings (higher is better). Values in parentheses indicate percentages change averaged over datasets relative to the baseline.

| Setting | $n_{\text{train}}$ | Baseline | CatBoost | Real-TabPFN | RealMLP | TabDPT | TabICL | TabPFN | XGB |
|---|---|---|---|---|---|---|---|---|---|
| ME | 100 | 0.884 | 0.835 (-6.1%) | 0.881 (-0.4%) | 0.861 (-3.0%) | 0.882 (-0.3%) | 0.878 (-0.8%) | 0.881 (-0.4%) | 0.845 (-5.1%) |
| | 200 | 0.886 | 0.851 (-4.5%) | 0.885 (-0.2%) | 0.873 (-1.6%) | 0.885 (-0.2%) | 0.884 (-0.3%) | 0.885 (-0.2%) | 0.852 (-4.5%) |
| | 300 | 0.886 | 0.863 (-3.0%) | 0.885 (-0.1%) | 0.877 (-1.1%) | 0.885 (-0.2%) | 0.885 (-0.1%) | 0.886 (-0.1%) | 0.860 (-3.5%) |
| | 400 | 0.887 | 0.866 (-2.7%) | 0.886 (-0.1%) | 0.879 (-1.0%) | 0.886 (-0.1%) | 0.886 (-0.1%) | 0.886 (-0.1%) | 0.862 (-3.3%) |
| | 500 | 0.888 | 0.870 (-2.2%) | 0.887 (-0.1%) | 0.882 (-0.7%) | 0.887 (-0.1%) | 0.887 (-0.1%) | 0.887 (-0.0%) | 0.866 (-2.9%) |
| ME + I | 100 | 0.850 | 0.808 (-5.4%) | 0.858 (+0.8%) | 0.826 (-3.0%) | 0.857 (+0.7%) | 0.853 (+0.1%) | 0.858 (+0.9%) | 0.812 (-5.0%) |
| | 200 | 0.862 | 0.833 (-3.8%) | 0.866 (+0.4%) | 0.849 (-1.8%) | 0.865 (+0.2%) | 0.866 (+0.3%) | 0.867 (+0.4%) | 0.829 (-4.4%) |
| | 300 | 0.866 | 0.843 (-3.0%) | 0.868 (+0.3%) | 0.858 (-1.0%) | 0.866 (+0.1%) | 0.868 (+0.3%) | 0.869 (+0.3%) | 0.839 (-3.6%) |
| | 400 | 0.869 | 0.848 (-2.7%) | 0.871 (+0.1%) | 0.861 (-1.0%) | 0.869 (-0.1%) | 0.871 (+0.1%) | 0.871 (+0.2%) | 0.843 (-3.4%) |
| | 500 | 0.871 | 0.853 (-2.3%) | 0.872 (+0.1%) | 0.865 (-0.7%) | 0.870 (-0.0%) | 0.872 (+0.1%) | 0.872 (+0.1%) | 0.847 (-3.1%) |
| ME + I + NL | 100 | 0.831 | 0.793 (-4.9%) | 0.842 (+1.1%) | 0.806 (-3.2%) | 0.841 (+1.1%) | 0.837 (+0.5%) | 0.842 (+1.2%) | 0.802 (-3.9%) |
| | 200 | 0.849 | 0.821 (-3.6%) | 0.857 (+0.9%) | 0.835 (-1.8%) | 0.854 (+0.5%) | 0.856 (+0.7%) | 0.858 (+0.9%) | 0.824 (-3.4%) |
| | 300 | 0.854 | 0.832 (-2.9%) | 0.861 (+0.7%) | 0.845 (-1.2%) | 0.858 (+0.3%) | 0.861 (+0.7%) | 0.862 (+0.8%) | 0.830 (-3.2%) |
| | 400 | 0.858 | 0.839 (-2.4%) | 0.864 (+0.6%) | 0.853 (-0.7%) | 0.861 (+0.3%) | 0.863 (+0.6%) | 0.864 (+0.7%) | 0.837 (-2.7%) |
| | 500 | 0.861 | 0.840 (-2.7%) | 0.865 (+0.4%) | 0.856 (-0.6%) | 0.862 (+0.1%) | 0.864 (+0.4%) | 0.865 (+0.5%) | 0.837 (-3.1%) |
| ME + I + NL + NI | 100 | 0.818 | 0.796 (-3.0%) | 0.842 (+2.8%) | 0.805 (-1.7%) | 0.841 (+2.7%) | 0.837 (+2.2%) | 0.842 (+2.8%) | 0.803 (-2.2%) |
| | 200 | 0.841 | 0.821 (-2.8%) | 0.857 (+1.8%) | 0.836 (-0.8%) | 0.854 (+1.4%) | 0.856 (+1.6%) | 0.858 (+1.8%) | 0.823 (-2.6%) |
| | 300 | 0.850 | 0.832 (-2.4%) | 0.861 (+1.3%) | 0.848 (-0.3%) | 0.858 (+0.9%) | 0.861 (+1.3%) | 0.862 (+1.3%) | 0.831 (-2.6%) |
| | 400 | 0.855 | 0.839 (-2.0%) | 0.864 (+1.0%) | 0.852 (-0.4%) | 0.861 (+0.7%) | 0.863 (+0.9%) | 0.864 (+1.0%) | 0.837 (-2.4%) |
| | 500 | 0.858 | 0.840 (-2.3%) | 0.865 (+0.8%) | 0.855 (-0.4%) | 0.862 (+0.4%) | 0.864 (+0.7%) | 0.865 (+0.8%) | 0.838 (-2.7%) |

The raw and relative Brier scores can be seen in Table 7.

*Table 7.* Brier scores across all experimental settings (lower is better). Values in parentheses indicate percentages change averaged over datasets relative to the baseline.

| Setting | $n_{\text{train}}$ | Baseline | CatBoost | Real-TabPFN | RealMLP | TabDPT | TabICL | TabPFN | XGB |
|---|---|---|---|---|---|---|---|---|---|
| ME | 100 | 0.115 | 0.168 (+76.8%) | 0.117 (+1.9%) | 0.138 (+25.8%) | 0.117 (+1.7%) | 0.118 (+2.9%) | 0.117 (+1.6%) | 0.159 (+54.0%) |
| | 200 | 0.111 | 0.147 (+59.9%) | 0.113 (+1.6%) | 0.126 (+18.6%) | 0.113 (+1.9%) | 0.112 (+0.2%) | 0.112 (+1.4%) | 0.146 (+47.1%) |
| | 300 | 0.109 | 0.133 (+41.1%) | 0.110 (+0.4%) | 0.121 (+18.2%) | 0.111 (+1.9%) | 0.110 (-0.4%) | 0.110 (+0.4%) | 0.138 (+40.8%) |
| | 400 | 0.109 | 0.131 (+37.2%) | 0.110 (+0.8%) | 0.119 (+15.1%) | 0.110 (+1.6%) | 0.109 (-0.1%) | 0.109 (+0.7%) | 0.134 (+37.1%) |
| | 500 | 0.108 | 0.127 (+33.5%) | 0.109 (+0.6%) | 0.117 (+13.7%) | 0.109 (+1.3%) | 0.109 (-0.1%) | 0.109 (+0.5%) | 0.129 (+32.3%) |
| ME + I | 100 | 0.142 | 0.190 (+41.2%) | 0.136 (-7.3%) | 0.166 (+17.6%) | 0.137 (-6.4%) | 0.139 (-4.9%) | 0.136 (-7.9%) | 0.188 (+35.5%) |
| | 200 | 0.133 | 0.160 (+27.0%) | 0.128 (-6.7%) | 0.146 (+9.7%) | 0.130 (-4.3%) | 0.128 (-7.1%) | 0.128 (-7.1%) | 0.170 (+31.5%) |
| | 300 | 0.128 | 0.149 (+22.7%) | 0.124 (-6.1%) | 0.138 (+7.9%) | 0.127 (-2.4%) | 0.124 (-6.3%) | 0.124 (-6.5%) | 0.157 (+28.9%) |
| | 400 | 0.126 | 0.145 (+20.5%) | 0.123 (-4.9%) | 0.135 (+8.4%) | 0.125 (-1.4%) | 0.123 (-5.2%) | 0.123 (-5.2%) | 0.152 (+27.0%) |
| | 500 | 0.124 | 0.141 (+18.5%) | 0.122 (-4.7%) | 0.132 (+6.2%) | 0.124 (-1.5%) | 0.122 (-4.8%) | 0.122 (-5.0%) | 0.148 (+25.3%) |
| ME + I + NL | 100 | 0.152 | 0.199 (+33.9%) | 0.143 (-8.2%) | 0.176 (+15.2%) | 0.145 (-6.2%) | 0.146 (-5.7%) | 0.142 (-8.5%) | 0.190 (+27.7%) |
| | 200 | 0.139 | 0.170 (+25.0%) | 0.132 (-7.8%) | 0.150 (+6.9%) | 0.135 (-4.5%) | 0.133 (-7.6%) | 0.132 (-8.1%) | 0.172 (+26.0%) |
| | 300 | 0.135 | 0.158 (+19.6%) | 0.128 (-7.9%) | 0.143 (+5.2%) | 0.132 (-4.4%) | 0.128 (-8.2%) | 0.128 (-8.2%) | 0.163 (+22.3%) |
| | 400 | 0.132 | 0.153 (+18.6%) | 0.126 (-6.9%) | 0.140 (+5.1%) | 0.129 (-3.5%) | 0.126 (-6.7%) | 0.126 (-7.2%) | 0.157 (+22.3%) |
| | 500 | 0.129 | 0.149 (+18.0%) | 0.125 (-6.1%) | 0.135 (+3.6%) | 0.128 (-2.4%) | 0.125 (-6.2%) | 0.125 (-6.5%) | 0.152 (+20.8%) |
| ME + I + NL + NI | 100 | 0.162 | 0.195 (+21.5%) | 0.143 (-14.4%) | 0.174 (+6.2%) | 0.145 (-12.6%) | 0.146 (-12.1%) | 0.142 (-14.6%) | 0.190 (+17.8%) |
| | 200 | 0.145 | 0.169 (+17.8%) | 0.132 (-12.1%) | 0.149 (+0.5%) | 0.135 (-9.1%) | 0.133 (-12.0%) | 0.132 (-12.4%) | 0.171 (+19.0%) |
| | 300 | 0.139 | 0.158 (+14.5%) | 0.128 (-11.0%) | 0.143 (+0.5%) | 0.132 (-7.7%) | 0.128 (-11.3%) | 0.128 (-11.3%) | 0.163 (+18.1%) |
| | 400 | 0.134 | 0.153 (+15.0%) | 0.126 (-9.5%) | 0.137 (+0.2%) | 0.129 (-6.2%) | 0.126 (-9.3%) | 0.126 (-9.8%) | 0.155 (+17.4%) |
| | 500 | 0.132 | 0.149 (+14.5%) | 0.125 (-8.4%) | 0.135 (+1.2%) | 0.128 (-4.9%) | 0.125 (-8.5%) | 0.125 (-8.8%) | 0.153 (+17.9%) |

The MSE values for the regression experiments are given in Table 8.

*Table 8.* Mean squared error (MSE) across all experimental settings (lower is better). Values in parentheses indicate percentages change averaged over datasets relative to the baseline

| Setting | $n_{\text{train}}$ | Baseline | CatBoost | Real-TabPFN | RealMLP | TabDPT | TabICL | TabPFN | XGB |
|---|---|---|---|---|---|---|---|---|---|
| ME | 100 | 27.013 | 98.105 (+254.7%) | 28.862 (+6.7%) | 39.136 (+43.5%) | 29.642 (+9.5%) | 28.786 (+6.5%) | 28.720 (+6.2%) | 66.351 (+141.6%) |
| | 200 | 25.884 | 68.554 (+163.7%) | 26.972 (+4.2%) | 33.294 (+28.8%) | 27.282 (+5.3%) | 26.631 (+2.9%) | 26.901 (+3.9%) | 52.779 (+103.5%) |
| | 300 | 25.593 | 56.808 (+121.3%) | 26.442 (+3.3%) | 31.671 (+23.9%) | 26.595 (+3.9%) | 26.044 (+1.8%) | 26.381 (+3.1%) | 48.139 (+87.6%) |
| | 400 | 25.512 | 49.925 (+94.5%) | 26.230 (+2.8%) | 30.208 (+18.0%) | 26.351 (+3.3%) | 25.878 (+1.4%) | 26.197 (+2.7%) | 45.679 (+78.2%) |
| | 500 | 25.225 | 44.947 (+77.4%) | 25.861 (+2.5%) | 29.318 (+16.1%) | 25.918 (+2.8%) | 25.516 (+1.2%) | 25.828 (+2.4%) | 43.245 (+70.8%) |
| ME + I | 100 | 45.342 | 306.763 (+408.2%) | 53.957 (+13.9%) | 88.414 (+74.0%) | 56.095 (+21.8%) | 60.280 (+28.5%) | 50.592 (+9.3%) | 194.978 (+237.3%) |
| | 200 | 30.252 | 213.665 (+525.3%) | 50.337 (+59.1%) | 57.516 (+79.1%) | 53.674 (+67.5%) | 43.145 (+36.8%) | 46.804 (+48.3%) | 136.823 (+307.0%) |
| | 300 | 27.939 | 166.687 (+456.0%) | 36.301 (+27.2%) | 47.386 (+64.9%) | 54.223 (+85.7%) | 38.096 (+33.1%) | 33.338 (+17.4%) | 112.330 (+278.5%) |
| | 400 | 27.126 | 142.260 (+394.2%) | 32.395 (+18.0%) | 41.621 (+49.8%) | 54.983 (+95.4%) | 34.385 (+24.9%) | 30.768 (+12.5%) | 100.191 (+251.6%) |
| | 500 | 26.534 | 125.074 (+347.1%) | 30.526 (+14.9%) | 39.638 (+46.6%) | 55.952 (+106.6%) | 32.467 (+21.6%) | 29.866 (+12.7%) | 91.540 (+229.6%) |
| ME + I + NL | 100 | 57.449 | 377.819 (+387.5%) | 68.691 (+17.4%) | 105.752 (+63.5%) | 64.708 (+15.4%) | 68.256 (+19.8%) | 65.625 (+13.3%) | 233.441 (+218.6%) |
| | 200 | 32.168 | 264.268 (+610.1%) | 59.640 (+65.9%) | 66.144 (+91.1%) | 64.185 (+84.3%) | 46.939 (+38.3%) | 52.786 (+49.2%) | 163.734 (+350.5%) |
| | 300 | 28.918 | 205.624 (+544.7%) | 37.597 (+26.9%) | 54.471 (+77.5%) | 65.865 (+114.6%) | 41.977 (+39.8%) | 35.809 (+21.4%) | 131.514 (+316.7%) |
| | 400 | 27.635 | 173.061 (+489.0%) | 36.912 (+31.2%) | 46.797 (+65.0%) | 67.254 (+132.2%) | 37.522 (+32.5%) | 35.201 (+25.2%) | 116.746 (+300.3%) |
| | 500 | 26.855 | 156.844 (+454.3%) | 32.688 (+21.2%) | 44.649 (+62.9%) | 69.540 (+152.6%) | 34.634 (+27.8%) | 30.791 (+14.4%) | 109.619 (+289.4%) |
| ME + I + NL + NI | 100 | 73.671 | 376.331 (+286.1%) | 68.691 (+3.0%) | 107.022 (+41.9%) | 64.748 (+0.9%) | 68.256 (+4.3%) | 65.625 (+0.4%) | 234.962 (+160.2%) |
| | 200 | 33.651 | 262.097 (+559.4%) | 59.640 (+59.1%) | 67.498 (+84.9%) | 64.203 (+75.4%) | 46.939 (+32.1%) | 52.786 (+42.7%) | 162.984 (+320.3%) |
| | 300 | 29.780 | 205.767 (+524.4%) | 37.597 (+23.1%) | 52.709 (+69.4%) | 65.948 (+107.2%) | 41.977 (+35.2%) | 35.809 (+17.8%) | 130.007 (+299.2%) |
| | 400 | 28.134 | 173.398 (+475.6%) | 36.912 (+28.8%) | 46.863 (+62.0%) | 67.247 (+127.0%) | 37.522 (+30.0%) | 35.201 (+23.0%) | 115.762 (+287.7%) |
| | 500 | 27.217 | 156.064 (+441.3%) | 32.688 (+19.8%) | 44.290 (+59.7%) | 69.619 (+148.2%) | 34.634 (+26.0%) | 30.791 (+13.0%) | 108.285 (+278.1%) |

The coverage rates, for the models which have uncertainty estimates, are given in Table 9.

*Table 9.* Empirical coverage values across experimental settings. The $\alpha$ is set to 0.05, resulting in a correct coverage rate of 0.95.

| Setting | $n_{\text{train}}$ | Baseline | Real-TabPFN | TabICL | TabPFN |
|---|---|---|---|---|---|
| ME | 100 | 0.951 | 0.938 | 0.948 | 0.941 |
| | 200 | 0.951 | 0.938 | 0.954 | 0.939 |
| | 300 | 0.949 | 0.935 | 0.955 | 0.937 |
| | 400 | 0.949 | 0.936 | 0.956 | 0.937 |
| | 500 | 0.952 | 0.938 | 0.958 | 0.939 |
| ME + I | 100 | 0.950 | 0.946 | 0.944 | 0.947 |
| | 200 | 0.950 | 0.943 | 0.953 | 0.944 |
| | 300 | 0.949 | 0.941 | 0.954 | 0.940 |
| | 400 | 0.949 | 0.941 | 0.956 | 0.940 |
| | 500 | 0.951 | 0.943 | 0.958 | 0.942 |
| ME + I + NL | 100 | 0.949 | 0.946 | 0.943 | 0.947 |
| | 200 | 0.950 | 0.944 | 0.952 | 0.944 |
| | 300 | 0.950 | 0.942 | 0.954 | 0.941 |
| | 400 | 0.950 | 0.942 | 0.957 | 0.941 |
| | 500 | 0.951 | 0.943 | 0.958 | 0.941 |
| ME + I + NL + NI | 100 | 0.950 | 0.946 | 0.943 | 0.947 |
| | 200 | 0.950 | 0.944 | 0.952 | 0.944 |
| | 300 | 0.949 | 0.942 | 0.954 | 0.941 |
| | 400 | 0.950 | 0.942 | 0.957 | 0.941 |
| | 500 | 0.952 | 0.943 | 0.958 | 0.941 |

Visualisation for the relative Brier scores, MSE, and absolute coverage rates can be seen in Figure 5/6/7.

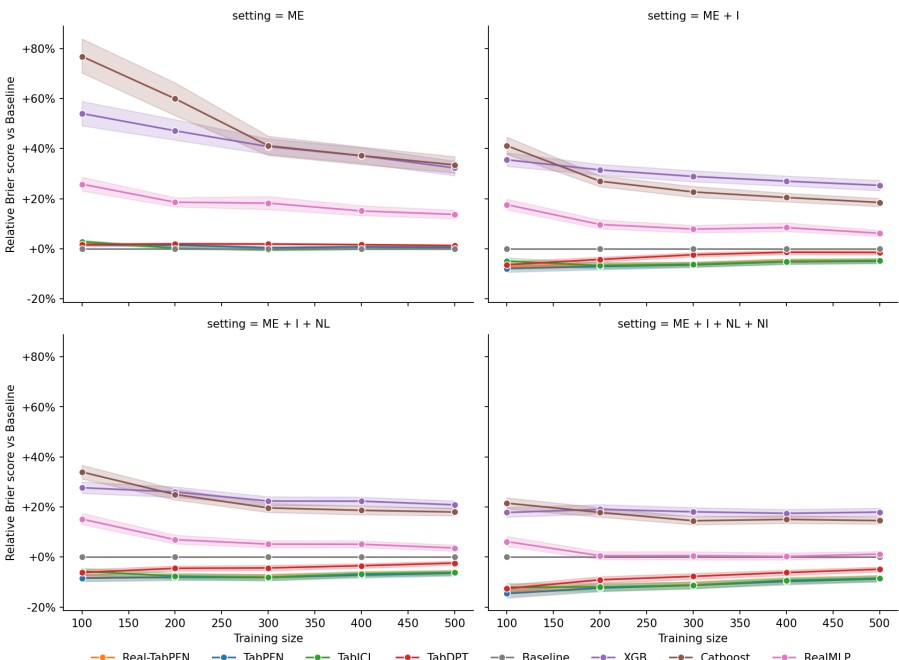

*Figure 5.* Relative Brier score to a correctly specified baseline model. Positive values indicate better performance. The error bars indicate the 95% confidence intervals around the mean. ME = Main effect, I = interactions, NL = Non-linear effects, NI = Non-important features. These are the aggregate results over the number of predictors.

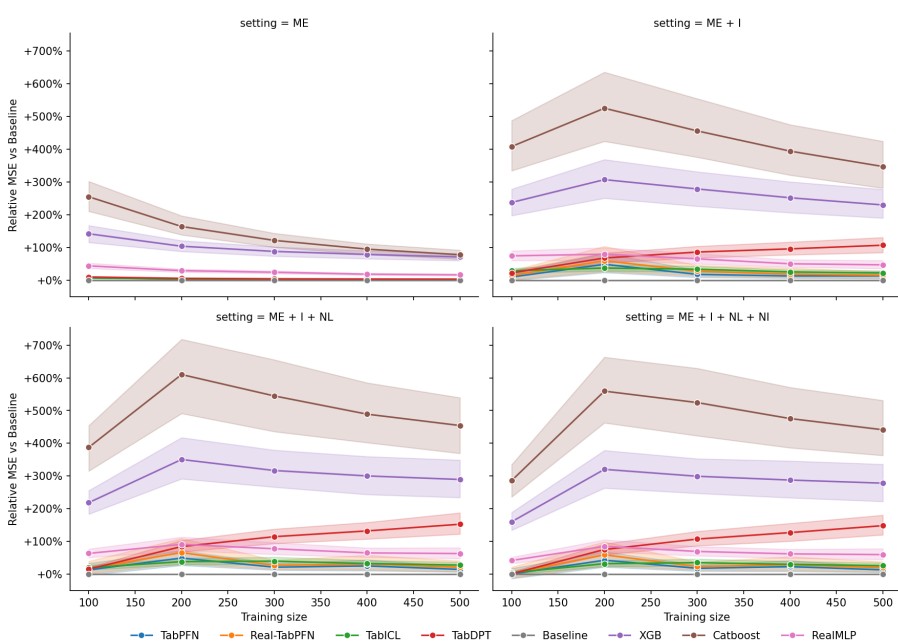

*Figure 6.* Relative MSE to a correctly specified baseline model. Negative values indicate better performance. The error bars indicate the 95% confidence intervals around the mean. ME = Main effect, I = interactions, NL = Non-linear effects, NI = Non-important features. These are the aggregate results over the number of predictors.

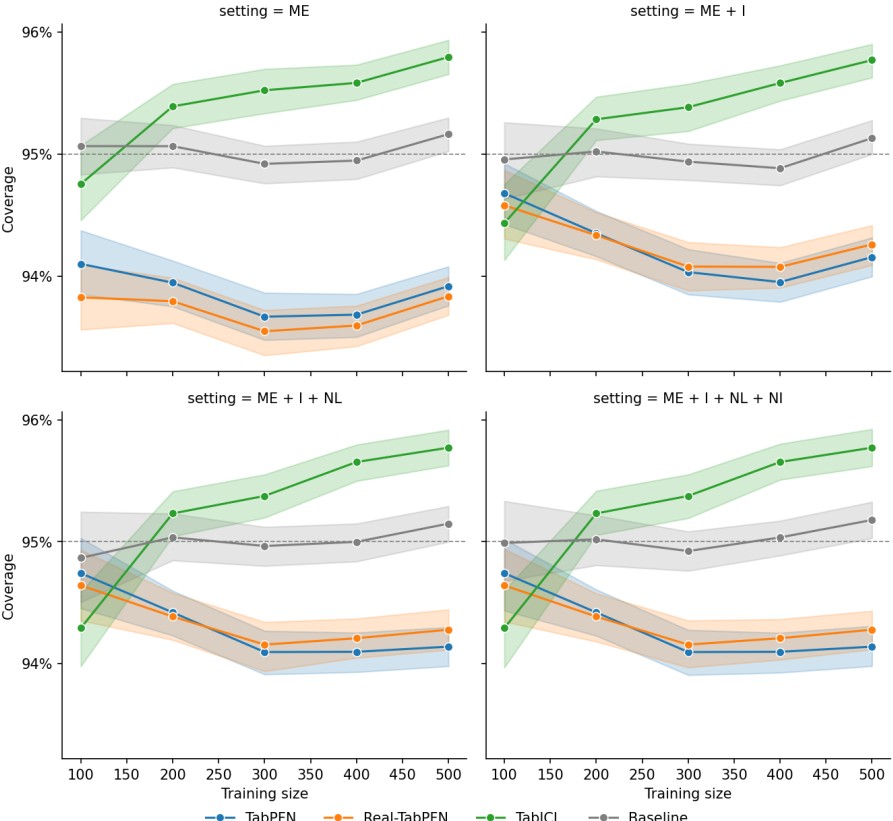

*Figure 7.* The absolute coverage rate for models with uncertainty estimates for the predictions. The dotted line indicates perfect coverage. The error bars indicate the 95% confidence intervals around the mean. ME = Main effect, I = interactions, NL = Non-linear effects, NI = Non-important features. These are the aggregate results over the number of predictors.

# D. Empirical experiments setup

For the empirical examples, the datasets from TabArena (Erickson et al., 2025) are used. Each dataset has multiple training/test folds, and the k-fold cross-validation is repeated a number of times per dataset. For each fold, we subsample a smaller training set and apply all models to this same subset. For the training cases, the subsample size is dictated by the experiment. If a dataset does not contain enough values for a specific experiment, it is not used in that setting. For the test, a maximum is set for the sample size due to computational constraints. For regression, random sampling is used to select cases, and for binary classification, stratification on the outcome is used.

The experimental setup is given in Table 10. For any given data set, two repeats are used; within a repeat, all folds are used. In every fold, 5 subsamples are taken for each training size.

*Table 10.* Empirical data subsample conditions

| Parameter | Conditions |
| --- | --- |
| Number of repeats | 2 |
| Number of subsamples for each fold | 5 |
| Train sample size | {100, 200, 300, 400, 500} |
| Maximum test size | 500 |

The data is cleaned based on the `AutoMLPipelineFeatureGenerator` function [4] from autogluon (Erickson et al., 2020). Due to the small sample sizes, some missing values occur when cleaning the categorical variables. Specifically, samples whose category has only one observation in the sample are set to missing values. All NaN are placed in an "other" category after cleaning. The Food Delivery Time dataset (363672) still had NaN values after this preprocessing and was not used.

The summary statistics for the datasets used in this study are given in Table 11 for the binary classification datasets, and Table 12 for the regression datasets.

*Table 11.* Summary binary classification datasets

| Name dataset | Sample size | Task id | N features | % cat features | Domain |
| --- | --- | --- | --- | --- | --- |
| Amazon_employee_access | 32769 | 363613 | 10 | 100.00 | business & marketing |
| bank-marketing | 45211 | 363618 | 14 | 57.14 | finance |
| Bank_Customer_Churn | 10000 | 363619 | 11 | 45.45 | finance |
| blood-transfusion-service-center | 748 | 363621 | 5 | 20.00 | medical & healthcare |
| churn | 5000 | 363623 | 20 | 25.00 | business & marketing |
| coil2000_insurance_policies | 9822 | 363624 | 86 | 4.65 | business & marketing |
| credit-g | 1000 | 363626 | 21 | 66.67 | finance |
| credit_card_clients_default | 30000 | 363627 | 24 | 16.67 | finance |
| diabetes | 768 | 363629 | 9 | 11.11 | medical & healthcare |
| E-CommereShippingData | 10999 | 363632 | 11 | 45.45 | business & marketing |
| hazelnut-spread-contaminant-detection | 2400 | 363674 | 31 | 3.23 | biology & life sciences |
| heloc | 10459 | 363676 | 24 | 4.17 | finance |
| in_vehicle_coupon_recommendation | 12684 | 363681 | 25 | 88.00 | business & marketing |
| Is-this-a-good-customer | 1723 | 363682 | 14 | 64.29 | business & marketing |
| NATICUSdroid | 7491 | 363689 | 87 | 100.00 | technology & internet |
| online_shoppers_intention | 12330 | 363691 | 18 | 44.44 | business & marketing |
| qsar-biodeg | 1054 | 363696 | 42 | 14.29 | biology & life sciences |
| seismic-bumps | 2584 | 363700 | 16 | 25.00 | environmental science & climate |
| taiwanese_bankruptcy_prediction | 6819 | 363706 | 95 | 1.05 | finance |

---

[4]The cleaning is based on this example from this TabArena github page.

*Table 12.* Summary regression datasets

| Name dataset | Task id | Sample size | N features | % cat features | Domain |
|---|---|---|---|---|---|
| airfoil_self_noise | 363612 | 1503 | 6 | 16.67 | physics & astronomy |
| Another-Dataset-on-used-Fiat-500 | 363615 | 1538 | 8 | 12.50 | technology & internet |
| concrete_compressive_strength | 363625 | 1030 | 9 | 0.00 | chemistry & material science |
| diamonds | 363631 | 53940 | 10 | 30.00 | business & marketing |
| healthcare_insurance_expenses | 363675 | 1338 | 7 | 42.86 | medical & healthcare |
| houses | 363678 | 20640 | 9 | 0.00 | business & marketing |
| miami_housing | 363686 | 13776 | 16 | 6.25 | business & marketing |
| physiochemical_protein | 363693 | 45730 | 10 | 0.00 | chemistry & material science |
| QSAR_fish_toxicity | 363698 | 907 | 7 | 0.00 | biology & life sciences |
| superconductivity | 363705 | 21263 | 82 | 0.00 | physics & astronomy |
| wine_quality | 363708 | 6497 | 13 | 7.69 | chemistry & material science |

# E. Empirical experiments additional results

The raw results from Figures 3 and 4 can be found in Tables 13/14.

*Table 13.* Average ranks by training sample size — Binary classification (lower is better)

| Method | AUC ↑ | | | | | Brier score ↓ | | | | |
|---|---|---|---|---|---|---|---|---|---|---|
| | 100 | 200 | 300 | 400 | 500 | 100 | 200 | 300 | 400 | 500 |
| Real-TabPFN | 2.89 | 2.72 | 2.73 | 2.90 | 2.73 | 2.27 | 2.26 | 2.22 | 2.21 | 2.11 |
| TabPFN | 3.25 | 2.93 | 2.98 | 3.22 | 3.19 | 2.58 | 2.44 | 2.43 | 2.56 | 2.43 |
| TabICL | 3.65 | 3.54 | 3.56 | 3.36 | 3.17 | 3.62 | 3.20 | 3.08 | 3.10 | 2.91 |
| TabDPT | 3.76 | 3.51 | 3.72 | 3.77 | 3.79 | 3.02 | 3.39 | 3.64 | 3.76 | 3.93 |
| CatBoost | 5.66 | 5.71 | 5.38 | 5.27 | 5.31 | 6.98 | 6.76 | 6.60 | 6.47 | 6.25 |
| XGB | 5.48 | 6.05 | 5.86 | 5.70 | 5.58 | 5.80 | 5.84 | 5.86 | 5.76 | 5.53 |
| RealMLP | 5.77 | 5.92 | 5.83 | 5.80 | 5.81 | 5.94 | 6.09 | 5.88 | 5.76 | 6.07 |
| Baseline | 5.55 | 5.61 | 5.96 | 5.98 | 6.41 | 5.79 | 6.02 | 6.29 | 6.38 | 6.77 |

*Table 14.* Average ranks by training sample size — Regression (lower is better)

| Method | MSE ↓ (rank) | | | | | Coverage (mean) | | | | |
|---|---|---|---|---|---|---|---|---|---|---|
| | 100 | 200 | 300 | 400 | 500 | 100 | 200 | 300 | 400 | 500 |
| TabICL | 2.80 | 2.24 | 1.86 | 1.71 | 1.74 | 0.937 | 0.947 | 0.950 | 0.949 | 0.950 |
| TabPFN | 2.48 | 2.53 | 2.60 | 2.77 | 2.71 | 0.946 | 0.944 | 0.944 | 0.942 | 0.942 |
| Real-TabPFN | 2.36 | 2.62 | 2.62 | 2.65 | 2.67 | 0.944 | 0.942 | 0.944 | 0.943 | 0.943 |
| TabDPT | 3.11 | 2.98 | 3.05 | 2.96 | 2.99 | —[a] | — | — | — | — |
| XGB | 5.50 | 5.76 | 5.86 | 5.88 | 6.05 | — | — | — | — | — |
| RealMLP | 5.84 | 5.84 | 6.06 | 6.30 | 6.36 | — | — | — | — | — |
| CatBoost | 7.51 | 7.25 | 6.74 | 6.37 | 6.05 | — | — | — | — | — |
| Baseline | 6.39 | 6.77 | 7.22 | 7.35 | 7.43 | 0.941 | 0.945 | 0.944 | 0.945 | 0.946 |

[a] Coverage is only available for models that output a predictive distribution.

The average rank, over the different training sample sizes, is given in Table 15/16. A Wilcoxon signed-rank test is used to assess if the models have significant differences in the rank. An alpha of 0.05 was used with a Bonferroni correction to correct for multiple testing.

The average AUC and MSE values by method and dataset can be seen in Table 17/ 18.

*Table 15.* Average rank across all binary classification experiments. Asterisks (*) indicate a significant difference compared to the previous method based on a Wilcoxon signed-rank test.

| Method | AUC Rank | Brier Score Rank |
|---|---|---|
| Real-TabPFN | 2.7 | 2.2* |
| TabICL | 2.7* | 2.7* |
| TabPFN | 3.2* | 2.8* |
| TabDPT | 3.4* | 3.3* |
| XGB | 5.7 | 6.1* |
| CatBoost | 5.7* | 6.5* |
| RealMLP | 6.3 | 6.4* |
| Logistic regression | 6.3 | 6.1 |

*Table 16.* Average rank across all regression experiments. Asterisks (*) indicate a significant difference compared to the previous method based on a Wilcoxon signed-rank test.

| Method | MSE Rank |
|---|---|
| TabICL | 2.0* |
| TabDPT | 2.7 |
| Real-TabPFN | 2.8* |
| TabPFN | 3.0* |
| XGB | 5.8* |
| RealMLP | 6.2* |
| CatBoost | 6.6* |
| Linear regression | 6.9 |

*Table 17.* The average AUC values by task id and method averaged over the subsamples. The standard deviation is in brackets.

| Task id | CatBoost | Logistic regression | Real-TabPFN | RealMLP | TabDPT | TabICL | TabPFN | XGB |
|---|---|---|---|---|---|---|---|---|
| 363613 | 0.522 (0.062) | 0.531 (0.055) | 0.541 (0.060) | 0.515 (0.064) | 0.542 (0.064) | 0.543 (0.062) | 0.539 (0.063) | 0.528 (0.058) |
| 363618 | 0.627 (0.055) | 0.619 (0.053) | 0.677 (0.050) | 0.616 (0.067) | 0.680 (0.050) | 0.672 (0.054) | 0.675 (0.053) | 0.624 (0.054) |
| 363619 | 0.783 (0.044) | 0.670 (0.039) | 0.837 (0.031) | 0.771 (0.060) | 0.823 (0.037) | 0.831 (0.033) | 0.836 (0.031) | 0.783 (0.043) |
| 363621 | 0.694 (0.043) | 0.749 (0.020) | 0.744 (0.035) | 0.725 (0.055) | 0.743 (0.036) | 0.739 (0.040) | 0.745 (0.035) | 0.681 (0.040) |
| 363623 | 0.860 (0.056) | 0.694 (0.048) | 0.880 (0.050) | 0.817 (0.070) | 0.858 (0.052) | 0.876 (0.050) | 0.876 (0.055) | 0.843 (0.074) |
| 363624 | 0.619 (0.078) | 0.630 (0.063) | 0.659 (0.066) | 0.580 (0.085) | 0.648 (0.068) | 0.645 (0.065) | 0.658 (0.069) | 0.602 (0.072) |
| 363626 | 0.718 (0.036) | 0.708 (0.033) | 0.750 (0.037) | 0.676 (0.044) | 0.738 (0.035) | 0.760 (0.038) | 0.744 (0.038) | 0.725 (0.038) |
| 363627 | 0.711 (0.038) | 0.642 (0.034) | 0.739 (0.033) | 0.689 (0.056) | 0.735 (0.032) | 0.741 (0.036) | 0.738 (0.032) | 0.703 (0.038) |
| 363629 | 0.807 (0.028) | 0.812 (0.029) | 0.833 (0.018) | 0.804 (0.036) | 0.833 (0.019) | 0.831 (0.019) | 0.832 (0.018) | 0.810 (0.022) |
| 363632 | 0.723 (0.030) | 0.712 (0.032) | 0.740 (0.026) | 0.703 (0.040) | 0.739 (0.025) | 0.740 (0.026) | 0.741 (0.026) | 0.725 (0.032) |
| 363674 | 0.916 (0.029) | 0.624 (0.070) | 0.956 (0.022) | 0.934 (0.025) | 0.957 (0.021) | 0.962 (0.022) | 0.953 (0.022) | 0.922 (0.026) |
| 363676 | 0.747 (0.035) | 0.732 (0.033) | 0.770 (0.029) | 0.737 (0.042) | 0.773 (0.027) | 0.769 (0.030) | 0.769 (0.028) | 0.748 (0.033) |
| 363681 | 0.630 (0.052) | 0.605 (0.042) | 0.679 (0.051) | 0.591 (0.041) | 0.632 (0.047) | 0.672 (0.046) | 0.673 (0.053) | 0.638 (0.049) |
| 363682 | 0.663 (0.052) | 0.671 (0.036) | 0.693 (0.045) | 0.633 (0.061) | 0.692 (0.039) | 0.692 (0.042) | 0.689 (0.044) | 0.649 (0.049) |
| 363689 | 0.958 (0.021) | 0.971 (0.007) | 0.972 (0.008) | 0.956 (0.016) | 0.970 (0.009) | 0.974 (0.008) | 0.972 (0.008) | 0.962 (0.012) |
| 363691 | 0.876 (0.031) | 0.778 (0.046) | 0.904 (0.021) | 0.824 (0.072) | 0.895 (0.020) | 0.903 (0.023) | 0.902 (0.021) | 0.877 (0.029) |
| 363696 | 0.892 (0.035) | 0.909 (0.020) | 0.918 (0.023) | 0.891 (0.027) | 0.918 (0.021) | 0.921 (0.022) | 0.916 (0.023) | 0.904 (0.027) |
| 363700 | 0.697 (0.058) | 0.350 (0.065) | 0.752 (0.049) | 0.664 (0.125) | 0.758 (0.048) | 0.758 (0.044) | 0.751 (0.052) | 0.677 (0.064) |
| 363706 | 0.852 (0.078) | 0.519 (0.077) | 0.896 (0.051) | 0.701 (0.176) | 0.893 (0.043) | 0.899 (0.051) | 0.895 (0.052) | 0.826 (0.106) |

*Table 18.* The average log MSE values by task id and method averaged over the subsamples. The standard deviation is in brackets.

| Task id | CatBoost | Logistic regression | Real-TabPFN | RealMLP | TabDPT | TabICL | TabPFN | XGB |
|---|---|---|---|---|---|---|---|---|
| 363612 | 2.25 (0.57) | 3.24 (0.09) | 1.54 (0.49) | 2.06 (0.54) | 1.56 (0.53) | 1.57 (0.54) | 1.54 (0.48) | 2.16 (0.49) |
| 363615 | 13.50 (0.16) | 13.33 (0.07) | 13.30 (0.07) | 13.47 (0.16) | 13.28 (0.08) | 13.27 (0.09) | 13.30 (0.07) | 13.45 (0.09) |
| 363625 | 4.02 (0.49) | 4.74 (0.08) | 3.34 (0.31) | 3.88 (0.27) | 3.41 (0.29) | 3.29 (0.31) | 3.34 (0.31) | 3.67 (0.31) |
| 363631 | 14.56 (0.59) | 15.19 (3.77) | 13.47 (0.70) | 14.33 (0.41) | 13.58 (0.38) | 13.62 (0.41) | 13.48 (0.72) | 14.18 (0.31) |
| 363675 | 17.59 (0.36) | 17.46 (0.15) | 16.95 (0.22) | 17.10 (0.28) | 16.98 (0.24) | 16.92 (0.21) | 16.94 (0.22) | 17.08 (0.16) |
| 363678 | -2.10 (0.26) | -2.09 (0.15) | -2.58 (0.18) | -2.06 (0.30) | -2.55 (0.17) | -2.60 (0.22) | -2.58 (0.20) | -2.15 (0.19) |
| 363686 | 24.49 (0.37) | 24.28 (0.20) | 24.22 (1.37) | 23.91 (0.40) | 23.73 (0.37) | 23.43 (0.37) | 24.34 (1.39) | 23.96 (0.38) |
| 363693 | 3.36 (0.12) | 3.43 (0.23) | 3.23 (0.12) | 3.42 (0.14) | 3.18 (0.13) | 3.19 (0.13) | 3.23 (0.12) | 3.37 (0.11) |
| 363698 | -0.08 (0.17) | -0.06 (0.09) | -0.20 (0.14) | -0.01 (0.28) | -0.21 (0.14) | -0.21 (0.15) | -0.19 (0.14) | -0.07 (0.14) |
| 363705 | 5.82 (0.28) | 8.34 (2.34) | 5.50 (0.21) | 5.87 (0.21) | 5.54 (0.21) | 5.48 (0.22) | 5.52 (0.21) | 5.67 (0.23) |
| 363708 | -0.56 (0.11) | -0.54 (0.11) | -0.61 (0.11) | -0.51 (0.13) | -0.65 (0.11) | -0.64 (0.12) | -0.61 (0.11) | -0.56 (0.11) |

