# OpenReview forum: "When Data Is Scarce: The Strength of the Prior in Tabular Foundation Models"
_ICML.cc/2026/Workshop/FMSD — FMSD @ ICML 2026 Poster_

### Official Review · Reviewer_2vsH · 2026-05-20
**Great work!**

**Rating:** 7
**Confidence:** 4

**Review:**

The paper "When data is scarce" studies the performance of TabPFN under extremely limited data. Concretely, it contains a study using synthetic data and a study using subsampled TabArena datasets. In the synthetic case, the paper compares TabPFN against models that generated the data, but where the parameters need to be fitted to the data. Both in cases with an uninformative and an informative prior TabPFN provides competitive performance (performs nearly equivalent or better), and for more complex models it was even able to improve over a correctly specified model in the extremely limited data regime (100 data points). Only for complex data generating models with a very strong prior TabPFN cannot reach the performance of the baseline model. For TabArena, as there is no prior, the baseline is a standard linear model (or logistic regression for classification), and requires more extensive hyperparameter optimization, and can thus not compete with TabPFN.

Then strengths of the paper are
1) demonstrating that TabPFN performs really well in the small data regime
2) that when a prior is available, a correct model can rival TabPFN
3) when an extremely good prior is available, a correct model might outperform TabPFN

The only weaknesses of this paper are the incorrect year of the original PFN paper (should be 2022 instead of 2024) and several references to arXiv instead of the published versions (e.g., the original PFN paper). Another "strawman" weakness is that there is no clear path to including priors into the inference mechanism of TabPFN when a strong prior is available. If the authors are looking for a way to extend the paper for a conference submission, they could think of even more complex models, that might lie outside of TabPFNs prior, e.g., very sparse data, and then run against a correctly specified model. An even more challenging follow-up might be to specify an SCM, sample data from it, and use the SCM structure as the correct model. Yet, I think the study is great for itself and should be presented at the workshop.

---

### Official Review · Reviewer_TWcM · 2026-05-20
**Evaluating PFNs on Data Limited Prediction Tasks**

**Rating:** 7
**Confidence:** 4

**Review:**

# Summary

This paper evaluates tabular PFNs (Real-TabPFN, TabPFN, TabICL, TabDPT) in data limited prediction tasks. Their experiments show that SOTA PFNs can match and outperform correctly specified models on synthetic tasks while also outperforming classical models (XGBoost, Catboost, MLP, LR) in real data tasks \- subsampled TabArena benchmark.

# Strengths

* The experimental setup is well designed to answer their primary research question on evaluating PFNs on limited data tasks.
* Consistent results showing that PFNs are strong predictors with limited sample size can be an interesting finding to researchers.

# Areas for Improvement \+ Detailed Comments

* It is stated that “Contrasting the parameter-level prior in parametric Bayesian models with the task-level prior in PFNs.” is a contribution of the paper. However, all of Section 3 is an explanation of existing knowledge in the PFN field and while important to include should not be considered as a contribution by the paper. I suggest you remove this statement from your contribution bulleted list, but keep Section 3\.
* While the analysis of your experimental results are adequate, I believe that the paper would benefit from a deeper analysis of your results. For example, how does training size affect performance and why were many models significantly worse at regression tasks than classification?
* While current experiments are satisfactory to answer the research question, readers would benefit from an extended experiment devising a more extensive scaling law on how training size impacts performance. Does using significantly larger sample sizes improve performance or is there diminishing returns at a certain size.
* The appendix is missing tables for model performance on real data aggregated by sample size. Include a table with the raw scores for Figures 3 and 4 beyond just average rank. Either include an aggregate score across tasks or include separate tables from a sample of the tasks.

---

### Official Review · Reviewer_9gm4 · 2026-05-22
**Novel research direction with initial promising results**

**Rating:** 6
**Confidence:** 4

**Review:**

This paper investigates the performance of tabular foundation models in a small sample regime (n < 500). Using synthetic toy models, the authors quantify how informative the prior must be for correctly specified models to match the predictive performance of PFNs. The paper shows that, in classification, PFNs match or slightly exceed correctly specified models with uninformative priors and can approach models with strong informative priors as n grows, while in regression they remain somewhat behind the ideal baseline yet still competitive. In addition to synthetic data, experiments have been conducted on real tasks from TabArena (subsampled to n < 500), showing that tabular foundations consistently outperform standard baselines (XGBoost, CatBoost, etc.) in the considered data regime.

The paper is well written and self-contained. The experiments are well defined and aligned with the claims of the paper.

Regarding the tabular data benchmarking, it appears to me that PFNs are already known to be more sample-efficient than their non-amortized equivalents. This has been established by the main PFN paper (Müller et al., 2022). Additionally, the TabPFN v1 paper (Hollmann et al., 2023) focuses on small tabular data. In this respect, these findings are not surprising to me.

However, the novelty of the paper lies in the controlled experiments since they give new insights into how to assess the importance of priors in PFNs. While the paper offers interesting initial results, it would be interesting to more explicitly connect the synthetic prior experiments to the specific, richer prior families used in TabPFN/TabICL to better understand why those particular priors work well. For example, would it make sense to consider a causal graph model instead of a simplistic parametric model?

Overall, I find the paper is a nice read with initial promising results.